# The Concordant Disruption of B7/CD28 Immune Regulators Predicts the Prognosis of Oral Carcinomas

**DOI:** 10.3390/ijms24065931

**Published:** 2023-03-21

**Authors:** Shi-Rou Chang, Chung-Hsien Chou, Chung-Ji Liu, Yu-Cheng Lin, Hsi-Feng Tu, Kuo-Wei Chang, Shu-Chun Lin

**Affiliations:** 1Institute of Oral Biology, College of Dentistry, National Yang Ming Chiao Tung University, Taipei 112304, Taiwan; 2Department of Dentistry, College of Dentistry, National Yang Ming Chiao Tung University, Taipei 112304, Taiwan; 3Department of Stomatology, Taipei Mackay Memorial Hospital, Taipei 104217, Taiwan; 4Department of Stomatology, Taipei Veterans General Hospital, Taipei 112201, Taiwan

**Keywords:** B7, CD28, head and neck cancer, immune modulator, lymph node, metastasis, oral cancer

## Abstract

Immune modulation is a critical factor in determining the survival of patients with malignancies, including those with oral squamous cell carcinoma (OSCC) and head and neck SCC (HNSCC). Immune escape or stimulation may be driven by the B7/CD28 family and other checkpoint molecules, forming ligand–receptor complexes with immune cells in the tumor microenvironment. Since the members of B7/CD28 can functionally compensate for or counteract each other, the concomitant disruption of multiple members of B7/CD28 in OSCC or HNSCC pathogenesis remains elusive. Transcriptome analysis was performed on 54 OSCC tumors and 28 paired normal oral tissue samples. Upregulation of CD80, CD86, PD-L1, PD-L2, CD276, VTCN1, and CTLA4 and downregulation of L-ICOS in OSCC relative to the control were noted. Concordance in the expression of CD80, CD86, PD-L1, PD-L2, and L-ICOS with CD28 members was observed across tumors. Lower ICOS expression indicated a worse prognosis in late-stage tumors. Moreover, tumors harboring higher PD-L1/ICOS, PD-L2/ICOS, or CD276/ICOS expression ratios had a worse prognosis. The survival of node-positive patients was further worsened in tumors exhibiting higher ratios between PD-L1, PD-L2, or CD276 and ICOS. Alterations in T cell, macrophage, myeloid dendritic cell, and mast cell populations in tumors relative to controls were found. Decreased memory B cells, CD8^+^ T cells, and Tregs, together with increased resting NK cells and M0 macrophages, occurred in tumors with a worse prognosis. This study confirmed frequent upregulation and eminent co-disruption of B7/CD28 members in OSCC tumors. The ratio between PD-L2 and ICOS is a promising survival predictor in node-positive HNSCC patients.

## 1. Introduction

Head and neck squamous cell carcinoma (HNSCC), occurring on the mucosal epithelium of the oral cavity, oropharynx, pharynx, and larynx, is the sixth most common malignancy in the world [1,2]. Chemical carcinogenic exposure (including smoking and drinking) and HPV infection are the main causes of HNSCC. Among Asians, betel chewing is an additional oral habit that predisposes individuals to a higher risk of oral squamous cell carcinoma (OSCC) [1]. Although the survival of HNSCC patients has been remarkably improved due to advances in treatment, resistance to drugs, relapse, and metastasis are still crucial factors affecting the outcomes of treatments.

The B7 immune checkpoint superfamily includes B7-1 (CD80), B7-2 (CD86), B7-H1 (PD-L1, CD274), PD-L2 (PDCD1LG2), B7-H3 (CD276), B7-H4 (B7-x, VTCN1), and the inducible costimulator ligand (L-ICOS, CD275), which act as ligands on the surface of tumor cells or antigen-presenting cells [3,4]. They couple with CD28 family members, including CD28, CTLA4, PD-1 (PDCD1), and ICOS (inducible costimulator CD278), or other receptors, to drive immune inhibition or stimulation [5,6]. CD28 and CTLA4 expressed on T lymphocytes share identical ligands, CD80 and CD86, on tumor cells or antigen-presenting cells [3,7]. CTLA4 upregulation is an important immunoinhibitory mechanism in HNSCC [8]. Although the scavenger effect of CTLA4 is due to its high affinity with CD80/CD86, which abrogates the immune stimulation roles of CD28, the feedback regulation and molecular interaction recently identified between CD28 and CTLA4 complicates their interplay in immune modulation [9,10]. Patients with HNSCC show abundant PD-1 expression in CD4^+^ T cells and CD8^+^ T cells, and both PD-L1 and PD-L2 react with PD-1 to mediate the signaling axis for immune suppression in the tumor microenvironment (TME). Studies have also demonstrated the differential affinity of PD-L1 and PD-L2 when binding to PD-1 [11,12].

L-ICOS/ICOS is the known complex that mediates immune stimulatory activity in the TME. High expression of ICOS is associated with improved survival of multiple malignancies, including HNSCC [13,14]. Although many interactive partners of CD276 have been found [15], the receptor of CD276 has not been identified thus far. Recent studies have shown that CD276 may exert inhibitory activities against immunity in HNSCC [5,16,17,18,19,20]. CD276 enables HNSCC stem cells to escape immune surveillance [18]. In addition, CD276 expression in the fibroblasts of TME is highly associated with the ferroptosis gene signature and poor HNSCC survival, suggesting its suppressive roles in tumor immunity [21]. Studies have also revealed the immune suppression activity of VTCN1 in tumors, and its expression defines the poor prognosis of HNSCC [22]. Apart from B7/CD28 superfamily regulators, perturbation of other immune checkpoint molecules is also involved in immune escape responses in the TME [4,23].

Antitumor immune responses are an emerging strategy in HNSCC therapy [2]. Reports have shown that approximately 15% of patients with metastatic HNSCC exhibit responses to anti-PD-1 therapy [3,24]. Moreover, neoadjuvant anti-PD-1/PD-L1 immunotherapy may have advantages in operable HNSCC according to histopathological evaluation [25]. PD-L2 targeting-based immune annotations may serve as alternative markers to facilitate anti-PD-1 therapy [26,27]. Due to the pluripotent roles of CD276 in HNSCC pathogenesis, and since CD276 is a predictor for the responses to immunotherapy in HNSCC [20], the combined efficacies of anti-PD-1 and anti-CD276 targeting have been tested in advanced HNSCC [28]. Many clinical trials attempting to abrogate or enrich the activity of B7/CD28 members are ongoing for HNSCC immunotherapy [23,24]. Although the functional activity and expression pattern of individual B7/CD28 members in HNSCC have been defined, since the functions of these molecules could be compensative, a comprehensive judgment of the prognostic value of B7/CD28 immune checkpoints in OSCC was needed. This study analyzed the transcription of B4/CD28 members in an OSCC tumor cohort to signify the immune landscape and prognostic value. Robust analysis of The Cancer Genome Atlas (TCGA) HNSCC dataset was also performed to validate the findings. We identified that the ratios of PD-L1, PD-L2, and CD276 transcripts in relation to ICOS transcripts are independent predictors of the outcomes in node-positive OSCC and HNSCC.

## 2. Results

### 2.1. Aberrances in the Expression of B7/CD28 Family Members in OSCC

Of the 54 OSCC patients enrolled in our study cohort, 96% were male, 55% were older than 60 years, 78% were stage IV, 74% were T4, and 44% were node-positive cases. Approximately 90% of patients had oral habits, including alcohol consumption, betel chewing, or tobacco smoking. During the follow-up period, 31% of patients died, and 28% of subjects experienced recurrence (Figure 1A). The transcriptome of OSCC samples and 28 paired normal controls was established by RNA sequencing analysis. Transcripts per kilobase of transcript per million mapped reads (TPM) of 11 B7/CD28 family members were retrieved from the transcriptome database. TPM values were also transformed into log_10_ to facilitate analysis. Compared to normal controls, upregulation of CD80, CD86, PD-L1, PD-L2, CD276, VTCN1, and CTLA4, and downregulation of L-ICOS were found in the tumors (Figure 1B). In the 28 paired tissue samples, upregulation of ICOS in OSCC was additionally observed (Figure 1C). The expression matrix based on the fragments per kilobase of transcript per million mapped reads upper quartile (FPKM-UQ) was downloaded from the HNSCC project in the TCGA Genomic Data Commons (GDC) data portal. Analysis of this cohort also revealed the upregulation of CD80, CD86, PD-L1, PD-L2, CD276, CTLA4, and ICOS and downregulation of L-ICOS in tumors in comparison to controls (Figure 1D,E), which was consistent with the findings in our OSCC cohort. The expression of VTCN1 in tumors and controls was not different in HNSCC samples in the TCGA cohort. The expression of CD28 and PD-1 in tumors and controls was not different in our OSCC cohort, nor in the TCGA HNSCC cohort.

### 2.2. Concordance in the Expression of B7/CD28 Family Members in OSCC

The heatmap of TPM in B7/CD28 members was plotted to show the generally high TPM of PD-L1, CD276, and CTLA4, as well as the low TPM of CD80 and VTCN1 in tumors (Figure 2A). Except for CD276 and VTCN1, the expression of the remaining B7 members was correlated with that of CD28 members (Figure 2B). Except for VTCN1, the expression of the remaining B7 members was generally intercorrelated (Figure 2C). The expression of members within the CD28 family was highly correlated (Figure 2D). The heatmaps of the node-negative OSCC subset, node-positive OSCC subset, and TCGA HNSCC samples are shown in Appendix A. The profiles of these two subsets generally simulated the profiles of all tumors.

### 2.3. Alterations in the Immune Cell Population in OSCC

The uploading of gene expression data into the CIBERSORT algorithm identified the differential immune cell annotation across controls and tumors in all OSCC samples (Figure 3A), paired samples (Figure 3B), all TCGA HNSCC samples (Appendix A), and the paired samples (Appendix A). Decreased CD8^+^ T cells, resting memory CD4^+^ T cells, monocytes, resting myeloid DCs, and activated mast cells, and increased M0 and M1 macrophages, were common events in tumors compared to normal tissues (Figure 3E). The analysis of node-positive (Figure 3C) and stage IV (Figure 3D) tumor subsets compared to their counterparts specified decreased myeloid DC resting and mast cell activation as common events in advanced OSCC subsets (Figure 3F). The analysis of TCGA HNSCC tumors (Appendix A) further revealed an additional increase in memory B cells and M0 macrophages in advanced HNSCC subsets (Figure 3F).

### 2.4. The Prognostic Implications of ICOS and CD276

The significance of B7/CD28 member expression for the prognosis of all node-negative and node-positive OSCC was analyzed. Lower ICOS expression was marginally associated with a worse prognosis in all OSCC (Figure 4A, Lt), and a worse prognosis in late-stage OSCC (Figure 4A, Rt). The analysis of other B7/CD28 members yielded no survival indication. Node-positive OSCC had a much worse prognosis than did patients without node involvement (Figure 4B, Lt), while lower ICOS expression in tumors further worsened the prognosis of node-positive OSCC (Figure 4B, Rt). CD276 was the only molecule among the B7 members whose higher expression defined a worse prognosis in TCGA HNSCC tumors (Figure 4C). However, higher expression of all CD28 members, whether immune inhibitory or stimulatory, defined better patient survival in all HNSCCs (Figure 4C). The survival implication matched the functional significance in CD276, CD28, and ICOS (Figure 4C). In the node-positive patient subset, the expression of CD276 and CD28 members also affected patient survival (Figure 4D, Appendix A). Interestingly, a higher CD276/ICOS ratio defined the worst prognosis in all HNSCC patients, and in the node-negative and node-positive patient subsets (Figure 4E).

### 2.5. The Prognostic Implications of PD-L1, PD-L2, and CD276 in Relation to CD28 Members

Although PD-L1 is a biomarker of the immunotherapeutic response and its expression in OSCC and HNSCC is remarkable, PD-L1 expression is not associated with tumor prognosis. We extensively examined the expression ratios of B7 members/CD28 members to designate survival prediction. Kaplan-Meier analysis revealed that the ratios of PD-L1 (Figure 5A,B) and PD-L2 (Figure 5C,D) to the expression of CD28 members enabled the discrimination of survival in a large proportion of all or node-positive OSCC patients, but not in patients without nodal involvement (Figure 6). The analysis of the TCGA HNSCC cohort yielded similar results (Figure 6, Appendix A). In addition to the CD276/ICOS ratio, the ratios of CD276 expression to the expression of other CD28 members also predicted survival (Figure 6 and Appendix A). The expression ratios of other B7 members over CD28 members were not associated with prognosis.

### 2.6. ICOS Expression and Ratios of PD-L1, PD-L2, and CD276 against ICOS Expression Were Independent Survival Predictors of Node-Positive HNSCC

Univariate logistic regression confirmed that CD276, ICOS, and the ratios of PD-L1, PD-L2, and CD276 expression/CD28 member expression were prognostic factors, as illustrated by Kaplan-Meier analysis in node-positive HNSCC. Multivariate modules, including logistic regression and Cox proportional hazards regression, specified that solitary ICOS, PD-L1/ICOS, PD-L2/ICOS, and CD276/ICOS were independent predictors of node-positive HNSCC (Table 1). The odds ratio and hazard ratio for PD-L2/ICOS were 3.4 and 13.0, respectively, for predicting mortality in node-positive HNSCC.

### 2.7. Alteration of the Immune Cell Population Associated with Worse Survival in Node-Positive Patients

CIBERSORT algorithms generally delineated attenuation in adaptive immune cells and enrichment of innate immune cells in node-positive HNSCC, harboring high ratios of PD-L1 or PD-L2/CD28 members (Figure 7A, Appendix A). However, in node-positive HNSCC with high ratios of CD276/CD28 members, the profile was complicated by enrichment in plasma cells and T CD4^+^ naïve cells, and attenuated M1 macrophages. The decreased B memory, CD8^+^ T, and Treg populations, together with the increased NK resting and M0 macrophage cell populations, were common in node-positive HNSCC with lower ICOS, or those cases with higher ratios of PD-L1, PD-L2, and CD276 relative to ICOS (Figure 7A,B).

## 3. Discussion

The expression states of B7 or CD28 family members in the TME of tumors, including HNSCC and OSCC, are known but still controversial. The pathogenetic characteristics of tumors associated with different etiological factors and the TME could be reasons for this discrepancy. The functional interplay between members may also result in differential profiles of B7/CD28 members in the TME. This study adopted an RNA sequencing strategy to gain contemporary insight into the expression levels of the members, in a cohort composed mainly of late-stage male OSCC patients with extensive oral habit exposure. Our approaches identified the upregulation of CD80, CD86, PD-L1, PD-L2, CD276, VTCN1, and CTLA4, and the downregulation of L-ICOS of the family members in the OSCC cohort. Although TPM was used to assess the transcripts in our cohort and KPFM-UQ was used for the TCGA cohort, the aberrations in these cohorts were nearly identical despite the clinical and etiological disparities [2]. Therefore, these B7/CD28 aberrations could be present in the human HNSCC or OSCC TME. This study also identified, for the first time, that the expression of B7/CD28 family members is correlated, except for CD276 and VTCN1. Correlations between the upregulation of CD80, CD86, and CD28 have been found in OSCC [3], and the lack of change in CD28 and PD-1 transcripts in our tumor samples, which might have been associated with the complexity in CTLA4 abundancy [9,10], requires further stratification. Since a correlation in the expression of multiple immune regulators has also been found in OSCC, targeting only one molecule may have limited therapeutic efficacy [24].

Changes in macrophages, T cells, myeloid DCs and other immune components in tumors compared to normal tissues have been found in TCGA HNSCC datasets [29]. These immune cell profiles also existed in our OSCC cohort. However, in contrast to the large decrease in M2 macrophages shown in TCGA HNSCC, there was only a marginal decreased in our cohort. The changes in the immune TME may suggest the feasibility of targeting immune components to intercept the oral neoplastic process. Our findings also showed that the resting myeloid DC and activated mast cell populations consistently decreased in both the occurrence stage and the progression stage of the tumors. However, B memory and M0 macrophages increased in HNSCC tumors, and resting NK cells increased in OSCC tumors during tumor progression. Although neck nodal metastasis is a key paradigm for accurate prognostic prediction [1], the functional significance underlying such discrepancies in the immune cell population during tumor metastasis requires further research.

In the TCGA HNSCC dataset, low expression of CD276 or high expression of CD28 family members in the TME were associated with better overall survival, while other markers were not prognostic factors. As CTLA4 and PD-1 are inhibitory immune regulators, the survival implications seem to conflict with their presumed functions. These preliminary findings in the tumor samples contradict the use of CTLA4 and PD-1 blockers as therapeutics [24]. The typing of immune infiltration cells or annotation of immune regulators in the HNSCC TME is necessary prior to therapy. ICOS may mediate the stimulation of tumor immunity, and its expression has been shown to be a survival predictor of HNSCC [13]. A recent study showed that ICOS and PD-1 label tumor-infiltrating T cells for neoantigen recognition in the TME [30]. In our OSCC cohort, only ICOS, among all B7/CD28 members, was a survival factor for late-stage tumors. Lower ICOS expression also had a worse prognosis in node-positive OSCC. The roles of CD276 as an immune-inhibitory factor in predicting HNSCC survival were much more clearly defined in this study [16,18,19,20]. Furthermore, a lower CD276/ICOS or higher ICOS was a marker predicting better survival of all patients and patients with nodal metastasis.

Although we found that PD-L1 and PD-L2 cannot be used for prognostic prediction for HNSCC or OSCC, the findings were similar to other studies [2]. Anti-PD-L1 therapy has been approved for neoadjuvant therapy of HNSCC and can improve the treatment outcomes, albeit with limited effects [3]. PD-L2 seems to be superior to PD-L1 in PD-1 binding [12], and it has been shown to be a prognostic factor of HNSCC in studies [26,27]. This study showed that the ratios of PD-L1 or PD-L2, normalized with ICOS and other CD28 members, also allowed for a good survival prediction for all cancers and HNSCC with nodal involvement in both cohorts using univariate modes. Tumor-infiltrating macrophages have been shown to be the dominant contributor of PD-L1 and other ligands in the HNSCC TME [31]. The interactive scenario between PD-L1, PD-L2, and CD276 ligands and the CD28 family receptors shown in this study, which suppresses the adaptive immune responses and exacerbates the innate immune population in the TME, may be an immune signature underlying disease outcomes.

Single ICOS expression and the ratios of PD-L1, PD-L2, or CD276 to ICOS were independent survival predictors of advanced HNSCC with cervical nodal metastasis in this study. The predictive power of the PD-L2/ICOS ratio was particularly eminent. The potent prediction power of the CD276/ICOS ratio in univariate modules was conspicuously reduced in multivariate modules after adjusting confounders. Decreased memory B cells, CD8^+^T cells and Tregs, and increased M0 and M2 macrophages in a fraction of metastatic tumors with a high PD-L2/ICOS ratio, may suggest future prospective therapeutic trials in this patient subpopulation. Despite tumor heterogeneity, the influences from unique immune components, HPV states, and protein data were not available in this approach, and the transcription investigation in this study identified the concordant disruption of the B7/CD28 checkpoint ligand–receptor interaction in HNSCC or OSCC tumors. Both PD-L2 and ICOS are important immune checkpoint regulators, but they are neglected in research. This study reinforced that PD-L2/ICOS expression in the TME could be a powerful biomarker for the prognosis of node-positive HNSCC patients.

## 4. Materials and Methods

### 4.1. Samples

Tumor samples were taken from 28 OSCC patients and their noncancerous matched normal tissue, and 26 unpaired OSCC tumors were collected at Taipei MacKay Memorial Hospital. The detailed information of this patient cohort has been described in Section 2.1. This study was approved by the ethics reviewing committees with an approval number of 18MMHIS187e. Written informed consent was obtained from patients prior to sampling.

### 4.2. RNA Extraction, RNA Sequencing, and Raw Data Filtering

Total RNA was extracted using the RNeasy kit (Qiagen, Hilden, Germany) and subjected to quality certification. After removal of rRNA, the remaining RNA was fragmented, modified, and subjected to RNA-Seq library construction using a SureSelect XT HS2 mRNA kit (Agilent, Santa Clara, CA, USA). Paired-end sequencing of the libraries was performed in a NovaSeq 6000 system (Illumina, San Diego, CA, USA) at Wegene Biotech (Taipei, Taiwan) [32]. Low-quality reads, or reads with contaminated adapters or excessive unrecognized bases, were excluded from the subsequent analysis.

### 4.3. Differential Expression Profiles

The clean reads of test genes in our sample cohort were subjected to bioinformatics analysis to achieve TPM values, which normalized the samples for cross comparison [33]. The values were also transformed into log_10_ in some analyses. The FPKM-UQ format of the HNSCC cohort was downloaded from the TCGA GDC data mode (http://cancergenome.nih.gov/) (accessed on 1 January 2023) to evaluate the differential expression of the tested genes [3,34,35]. The CIBERSORT algorithm (https://cibersortx.stanford.edu/) (accessed on 1 January 2023) was used to estimate the abundance of 22 tumor-infiltrating immune cells in the complex TME of samples using the input gene expression data [36].

### 4.4. Statistics

Data were shown as their original values or mean ± standard error (SE). Patients were divided into high and low expression groups according to the median value of the variants. Heatmaps illustrated the differences in clinical parameters and matrix values. The Mann-Whitney test, Wilcoxon’s signed rank test, *t*-tests, and linear correlation analysis were performed. Kaplan-Meier survival analysis was used to assess the association between gene expression states and overall survival. The prognostic signatures were also analyzed using univariate and multivariate logistic regression and the Cox proportional hazards regression module. Statistical analyses were performed using Prism 9.0 software (GraphPad, San Diego, CA, USA). ns, not significant. *, ** and *** represent *p* < 0.05, *p* < 0.01 and *p* < 0.001, respectively.

## Figures and Tables

**Figure 1 ijms-24-05931-f001:**
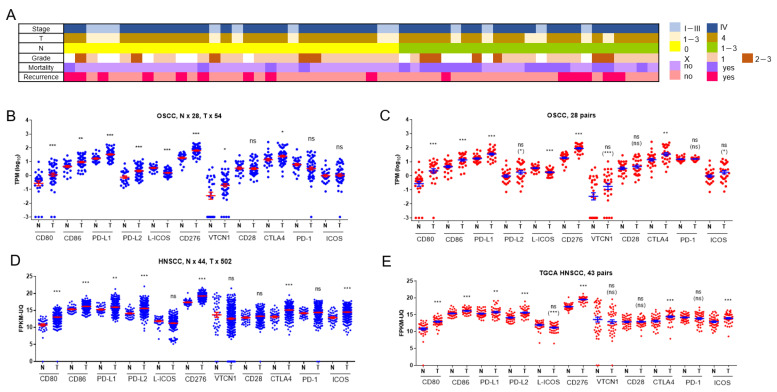
Clinicopathological parameters and B7/CD28 transcripts. (**A**) Clinicopathological parameters, including the clinical stage, tumor size, cervical node metastasis, differentiation grade of the tumors, and the mortality and recurrence states during the follow-up period of our OSCC patients. The colored boxes designate the clinicopathological status. (**B**–**E**). Scatter dot plots. (**B**,**C**) Log_10_ (TPM) of B7/CD28 members in our OSCC and control. (**D**,**E**) FKPM-UQ of B7/CD28 members in the TCGA GDC HNSCC cohort. The mean and SE of each group are also incorporated in the dot plots. (**B**,**D**) Mann-Whitney test of all samples. (**C**,**E**) Mann-Whitney test and Wilcoxon signed rank test of paired samples. The comparisons showing no statistically significant difference using the Mann-Whitney test were re-analyzed with the Wilcoxon signed rank test, and the statistical values are shown in the parentheses. N, normal; T, tumor. ns, not significant. *, ** and *** represent *p* < 0.05, *p* < 0.01 and *p* < 0.001, respectively.

**Figure 2 ijms-24-05931-f002:**
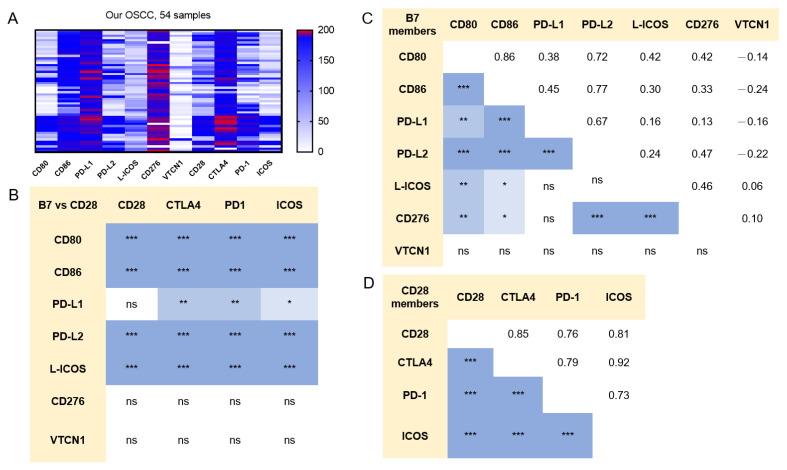
Correlation of B7/CD28 member expression in our OSCC cohort. (**A**) Heatmap to illustrate the TPM of B7/CD28 members in tumors. Gradient bar, TPM. (**B**) Correlation between B7 members and CD28 members. (**C**) Correlation among B7 members. (**D**) Correlation among CD28 members. Numbers in (**C**,**D**), γ values. The differential densities of blue color in boxes in (**B**–**D**) indicate the different degrees of correlation. ns, not significant. *, ** and *** represent *p* < 0.05, *p* < 0.01 and *p* < 0.001, respectively.

**Figure 3 ijms-24-05931-f003:**
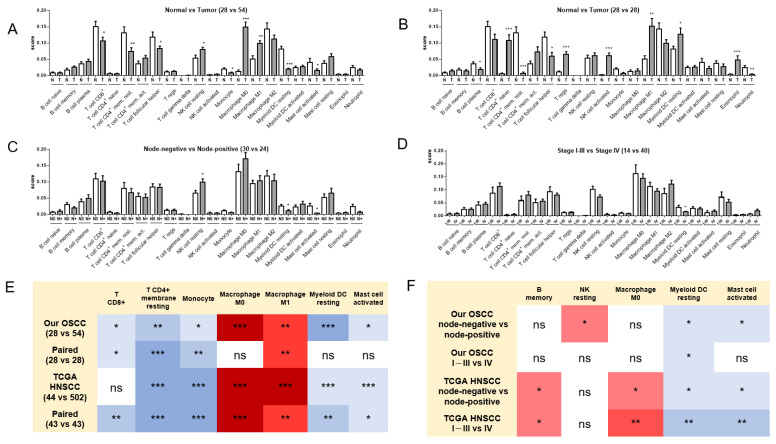
CIBERSORT algorithms to delineate the immune cell infiltration in our OSCC samples. (**A**–**D**) Analysis plots. (**A**) Normal vs. tumor. (**B**) Paired normal vs. tumor. (**C**) Node-negative tumor vs. node-positive tumor. (**D**) Stage I–III tumor vs. stage IV tumor. N, normal; T, tumor, N0, node-negative tumor; N+, node-positive tumor. (**E**) Summary of the results of unpaired *t*-tests and paired *t*-tests in our OSCC and TCGA HNSCC. Only alterations present in both cohorts are organized in this diagram. The first number within parenthesis is the number of normal samples. The second number within parenthesis is the number of tumor samples. (**F**) Comparison between node-negative and node-positive, and between stage I–III and stage IV tumors, in our OSCC cohort and TCGA HNSCC cohort. In (**E**,**F**), the differential densities of blue color in the boxes indicate the different degrees of downregulation, while the differential densities of red color in the boxes indicate the different degrees of upregulation. ns, not significant. *, ** and *** represent *p* < 0.05, *p* < 0.01 and *p* < 0.001, respectively.

**Figure 4 ijms-24-05931-f004:**
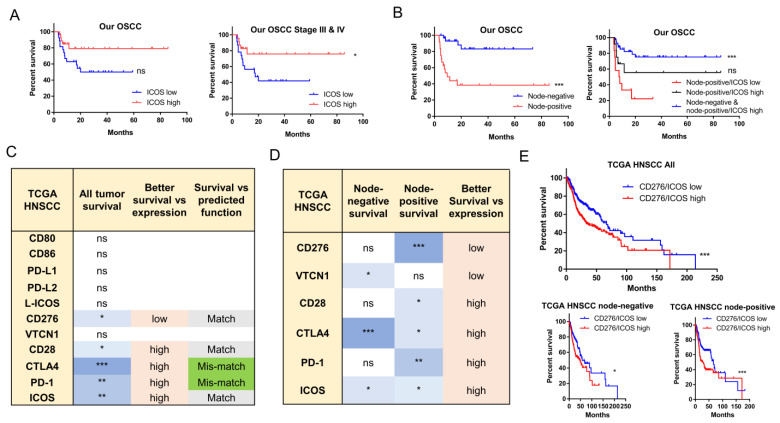
Survival states analyzed with the Kaplan-Meier mode. (**A**) According to ICOS expression. Lt, All OSCC tumors; Rt, late stage OSCC tumors. (**B**) All OSCC tumors. Lt, according to node involvement. Rt, according to node involvement and ICOS expression. (**C**–**E**) TCGA HNSCC tumors. (**C**) As related to B7/CD28 members. (**D**) As related to selected B7/CD28 members and nodal status. The detailed analysis integrates the supplementary results. (**E**) According to the CD276/ICOS ratio in all tumors (Lt), node-negative tumors (middle) and node-positive (Rt) tumors. In (**C**,**D**), the different densities of blue color in the boxes indicate the different degrees of correlation. ns, not significant. *, ** and *** represent *p* < 0.05, *p* < 0.01 and *p* < 0.001, respectively.

**Figure 5 ijms-24-05931-f005:**
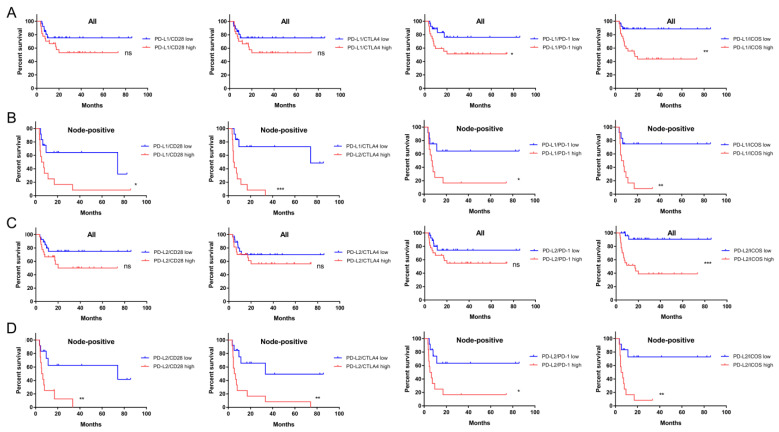
Survival analysis according to PD-L1 or PD-L2 in relation to CD28 member ratios in our OSCC cohort. (**A**,**B**) PD-L1/CD28 members. (**C**,**D**) PD-L2/CD28 members. (**A**,**C**) All tumors. (**B**,**D**) Node-positive tumors. Lt, CD28. Lt Middle, CTLA4, Rt Middle, PD-1, Rt, ICOS. ns, not significant. *, ** and *** represent *p* < 0.05, *p* < 0.01 and *p* < 0.001, respectively.

**Figure 6 ijms-24-05931-f006:**
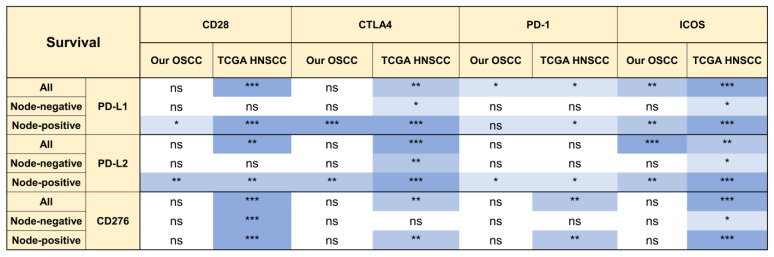
Summary of survival analysis according to PD-L1, PD-L2, or CD276/CD28 members in our OSCC and TCGA HNSCC. ns, not significant. *, ** and *** represent *p* < 0.05, *p* < 0.01 and *p* < 0.001, respectively.

**Figure 7 ijms-24-05931-f007:**
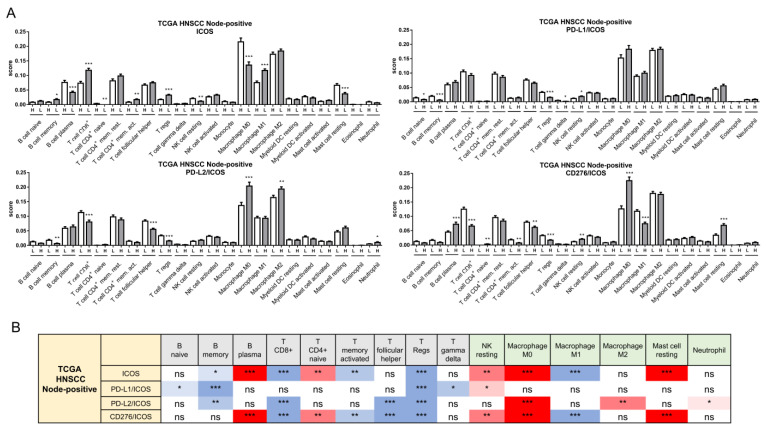
CIBERSORT algorithms to delineate the immune cell infiltration in TCGA HNSCC node-positive tumor subsets. (**A**) Analysis of individual ICOS (Upper Lt), PD-L1/ICOS (Upper Rt), PD-L2/ICOS (Lower Lt), and CD276/ICOS (Lower Rt), respectively. (**B**) Summary of immune cell infiltration according to ICOS, PD-L1/ICOS, PD-L2/ICOS, and CD276/ICOS, respectively. Grey boxes, adaptive immune cells; green boxes, innate immune cells. The differential densities of blue colors in boxes indicate the different degrees in decreasing of cell population, while the differential densities of red colors in boxes indicate the different degrees in increasing of cell population. ns, not significant. *, ** and *** represent *p* < 0.05, *p* < 0.01 and *p* < 0.001, respectively.

**Table 1 ijms-24-05931-t001:** Univariate and multivariate analysis for prognostic predictors in HNSCC with nodal metastasis.

TCGA HNSCC Node-Positive	Logistic Regression (Univariate)			Logistic Regression (Multivariate)			Cox Proportional Hazards Regression	
	OR	95% CI	*p*	OR	95% CI	*p*	HR	95% CI
CD276	1.33	1.02 to 1.74	0.024 *	1.39	1.06 to 1.84	ns	1.30	1.06 to 1.60
ICOS	0.79	0.67 to 0.93	0.018 *	0.77	0.65 to 0.91	0.0018 **	0.85	0.76 to 0.95
PD-L1/CD28	11.61	2.14 to 67	0.003 **	0.18	0.01 to 5.61	ns	0.28	0.03 to 2.99
PD-L1/CTLA4	173	16.87 to 2194	<0.0001 ***	116	0.33 to 83,302	ns	2.30	0.05 to 100
PD-L1/PD-1	69	8.21 to 678	0.0002 ***	11.45	0.32 to 424	ns	10.73	1.01 to 111
PD-L1/ICOS	76	9.02 to 739	0.0001 ***	1.05	0.00 to 594	<0.0001 ***	3.33	0.04 to 226
PD-L2/CD28	7.78	1.45 to 45.51	0.012 *	0.16	0.00 to 5.21	ns	0.20	0.02 to 2.00
PD-L2/CTLA4	114	10.88 to 1547	<0.0001 ***	110	0.27 to 86,428	ns	1.49	0.04 to 63
PD-L2/PD-1	23.33	3.41 to 177	0.001 ***	2.26	0.06 to 76	ns	6.04	0.62 to 57
PD-L2/ICOS	80	8.05 to 961	0.0004 ***	3.40	0.00 to 2670	<0.0001 ***	13.00	0.12 to 996
CD276/CD28	5.44	1.45 to 21.66	0.007 **	0.51	0.03 to 8.11	ns	0.56	0.08 to 3.81
CD276/CTLA4	41.50	6.70 to 304	<0.0001 ***	298	1.88 to 95,050	ns	3.56	0.22 to 58
CD276/PD-1	8.98	2.15 to 40.14	0.003 **	0.77	0.04 to 12.82	ns	3.39	0.54 to 21.00
CD276/ICOS	17.87	3.63 to 103	0.001 ***	0.33	0.00 to 52	0.0001 ***	1.24	0.04 to 28.76

ns, not significant. *, ** and *** represent *p* < 0.05, *p* < 0.01 and *p* < 0.001, respectively.

## Data Availability

Not applicable.

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
