# Peer review of "The Concordant Disruption of B7/CD28 Immune Regulators Predicts the Prognosis of Oral Carcinomas"

_ijms, 2023, doi:10.3390/ijms24065931_

Round 1

Reviewer 1 Report

Shi-Rou Chang et al. showed upregulation of CD80, CD86, PD-L1, PD-L2, CD276, VTCN1 and CTLA4 and downregulation of L-ICOS in OSCC based on human OSCC tumor and TCGA dataset. The author further found the alterations of multiple subtypes of immune cells. The prognostic analysis provided many interesting results. Higher PD-L1/ICOS, PD-L2/ICOS or CD276/ICOS expression ratios exhibited worse prognoses. And, the survival of node-positive patients was worsened in tumors exhibiting higher ratios between PD-L1, PD-L2 or CD276 and ICOS. In the all, In the all, the results of the study are well described. However, some aspects should be addressed.

1In the abstract, the descriptive logic of the results needs to be improved. eg.” Memory B cells, CD8+ T cells and Tregs were decreased and resting NK cells and M0 macrophages were increased in this subset of HNSCC samples”. The immune cell changes should not be described in two parts.

2.  In the result, the author should describe the method of quantitative analysis in the beginning of the part.

3.  Did the author perform paired t test in 28 pairs OSCC?

4.  The description of material and method is simple. Statistical and analytical methods should be written in more detail.

Author Response

Q1. In the abstract, the descriptive logic of the results needs to be improved.
The immune cell changes should not be described in two parts.

A1. Thank you for your excellent comments. We have merged the descriptions of changes regarding immune cell in the abstract. Please refer to lines 30 - 33 in the revised abstract.

Q2.  In the result, the author should describe the method of quantitative analysis in the beginning of the part.

A2. We acknowledge your opinions. We have followed your suggestions and incorporated essential definition, quantitative methods of transcripts and the algorithmic approaches being used for data analysis in the beginning part of the results to facilitate reading. Please refer to lines 101 – 104, and 108 - 110 in the revised manuscript.    

Q3.  Did the author perform paired t test in 28 pairs OSCC?

A3. Thank you. We have done Wilcoxon signed rank test (non-parametric paired test) in Fig. 1C and Fig. 1E. The statistical values are incorporated in the parenthesis of the revised diagrams. The paired tests validate the statistically significant results being shown by unpaired tests. Besides, the original comparisons in several normal/tumor pairs, showing no differences or only marginal discrepancies, turn to be significantly different due to the paring effects. This is particularly eminent in the paired tissues (28 pairs) of our sample cohort. 

Q4.  The description of material and method is simple. Statistical and analytical methods should be written in more detail.

A4. Thank you for your comments. During revision we have incorporated more detailed descriptions in material and method section, which include the quality control of RNA, a reference citing our previous RNA-Seq methodology, the FPKM-UQ quantitation system and its associated reference, the websites for assessing algorithms and the software for the statistical analysis. Please refer to lines 336, 340, 341, 350 - 355, 357, 364 and 365 for details.

Reviewer 2 Report

Dear Authors, 

first of all - congratulations for your great effort and interesting topic of the paper. I hope that my hints will be helpful and your final paper will be of great value to the scientific community. 

- Despite the well-rooted habit of using abbreviations in Medicine sometimes too many such shortcuts make it more difficult to understand instead of facilitating the topic. It might be worth considering  to only use well-known shortcuts, but not to create too many new ones.

- There are several new papers touching the topic of immunotherapy and checkpoint molecules, please update your literature review. Many important papers are missing or outdated. As a result, several information in the text need to be revised and corrected, for example information about the CD276 molecule comes from the paper from 2016, many new findings have been described until today, including the pair for this checkpoint molecule. Please, revise your manuscript and check every information given. 

- Several structural changes are necessary. For example 2.1. beginning is more appropriate for the materials and methods than results; it is clearly the cohort description. 

- Figures, especially figure 1 and 3. have too small letters, incomprehensible and difficult to read. Figure description is absolutely unacceptable - there is no explanation of what can be observed on the graphs above. 

- Some minor language mistakes are also present in the text, especially grammar mistakes.

- There are some typos and punctuation errors, e.g. double space or double dot. Please review your manuscript carefully. 

Author Response

Q1. Despite the well-rooted habit of using abbreviations in Medicine sometimes too many such shortcuts make it more difficult to understand instead of facilitating the topic. It might be worth considering to only use well-known shortcuts, but not to create too many new ones.

A1. We appreciate your comments. During revision, we have eliminated the non-official abbreviations N0 and N+. The term N0 has been replaced by “node-negative” or “tumor without nodal involvement” in the revised text. The term N+ has been replaced by “node-positive” or “tumor with cervical nodal metastasis” all over the manuscript. N0 and N+ have been only minimally maintained in two diagrams (Fig. 3C and Fig. S2C) to illustrate the subsets. Thank you again for your precious opinions. 

Q2. There are several new papers touching the topic of immunotherapy and checkpoint molecules, please update your literature review. Many important papers are missing or outdated. As a result, several information in the text need to be revised and corrected, for example information about the CD276 molecule comes from the paper from 2016, many new findings have been described until today, including the pair for this checkpoint molecule. Please, revise your manuscript and check every information given.

A2. Thank you for your excellent opinions. We agree with your opinions and would like to address your queries in the following.

  1. CD276: We have vigorously reviewed the more recently published papers (2018-2023) regarding CD276 issue in HNSCC. The key findings in those recent reports revealed the pluripotent roles of CD276 in carcinogenesis and immune escape, which are in concordance with your concerns and supportive to the results of this study. Therefore, the CD276 part in introduction section has been re-written. Please refer to descriptions in lines 66 - 71, which address the versatile functions of CD276 in HNSCC oncogenicity, stemness and the immune escape by modulating immune microenvironment. Please also refer to descriptions in lines 80 - 84, which signify the potential efficacy of CD276 targeting in HNSCC therapy.
  2. CD28/CTLA4: These two receptors have been known competitive to each other due to the sharing of same CD80/CD86 ligands for years. The opposite immune modulation activities of CD28 (stimulatory) and CTLA4 (inhibitory) have also been widely reported. However, recent studies have revealed the complicated regulation circuits lying between CD28 and CTLA4. Besides, the differential modification driving from CD80/CD86 associated molecular ecosystem and cytokines would also affect the homeostasis of stimulatory/inhibitory activities elicited by CD28/CTLA4. Therefore, we have briefly incorporated the updated findings about CD28/CTLA4 in lines 55 – 59 of the revised manuscript.             
  3. We have carefully checked the descriptions on PD-L1/PD-L2 and PD-1, and ICOS/L-ICOS, and revised some wordings to eliminate the misunderstanding. 
  4. A number of 8 papers, mostly published after 2020, related to HNSCC immune modulation and immune therapy, which support the rationale or objective of this research are cited in the revised manuscript.   

Q3. Several structural changes are necessary. For example 2.1. beginning is more appropriate for the materials and methods than results; it is clearly the cohort description.

A3. Thank you for your suggestions. We agree that the cohort description should be documented in the beginning of material and method section. But since the manuscript template of IJMS follows the Introduction-Results-Discussion-Methods consequence, we incorporate the clinicopathological information in section 2.1 and Fig. 1A in this paper. To follow the structure of IJMS journal, we apologize that we put the cohort information in section 2.1 as the beginning. 

Q4. Figures, especially figure 1 and 3. have too small letters, incomprehensible and difficult to read. Figure description is absolutely unacceptable - there is no explanation of what can be observed on the graphs above.

A4. Thank you. We apologize for our inappropriateness. During revision, efforts have been laid to improve this mistake.

  1. The symbols and letters in the diagrams and the size of the pictures are enlarged as possible as we can.
  2. High resolution images will be re-submitted to assure the clarity.
  3. The legends of Fig. 1 and Fig. 3 are re-written to encompass sufficient explanation of symbols and color gradients being used to interpretate data. 
  4. Likewise, critical explanations are also added to legends of other graphs, such as Fig 7B and Fig. S9, to enrich the interpretation. 

Q5. Some minor language mistakes are also present in the text, especially grammar mistakes.

A5. Thank you for your critiques. This revised manuscript has been subjected to language editing affiliated to the publisher for proofreading prior to re-submission.

Q6. There are some typos and punctuation errors, e.g. double space or double dot. Please review your manuscript carefully. 

A6. We have carefully corrected typos (such as FKPM), inconsistent punctation (such as CD4+ T cell), missing of period in author name initial and other errors during revision. We appreciate your kindness in pointing out these errors.

Round 2

Reviewer 2 Report

Dear Authors, 

thank you for your time and corrections, 

good luck in your further actions, 

P.